# LEARNING REPRESENTATIONS OF CATEGORICAL FEATURE COMBINATIONS VIA SELF-ATTENTION

## ABSTRACT

Self-attention has been widely used to model the sequential data and achieved remarkable results in many applications. Although it can be used to model dependencies without regard to positions of sequences, self-attention is seldom applied to non-sequential data. In this work, we propose to learn representations of multi-field categorical data in prediction tasks via self-attention mechanism, where features are orderless but have intrinsic relations over different fields. In most current DNN based models, feature embeddings are simply concatenated for further processing by networks. Instead, by applying self-attention to transform the embeddings, we are able to relate features in different fields and automatically learn representations of their combinations, which are known as the factors of many prevailing linear models. To further improve the effect of feature combination mining, we modify the original self-attention structure by restricting the similarity weight to have at most $k$ non-zero values, which additionally regularizes the model. We experimentally evaluate the effectiveness of our self-attention model on non-sequential data. Across two click through rate prediction benchmark datasets, i.e., Cretio and Avazu, our model with top-$k$ restricted self-attention achieves the state-of-the-art performance. Compared with the vanilla MLP, the gain by adding self-attention is significantly larger than that by modifying the network structures, which most current works focus on.

## 1    INTRODUCTION

Self-attention, as a special attention mechanism, has been widely used to model the sequential data and achieved very remarkable results in many applications, e.g., neural machine translation (Vaswani et al., 2017), sentiment analysis (Lin et al., 2017b), and reading comprehension (Wang et al., 2017b). Different from recurrent and convolutional neural networks (Gehring et al., 2017), self-attention models dependencies without regard to the positions of sequences. In order to capture the sequential behavior, such attention mechanism is often used in conjugate with recurrent neural networks (Lin et al., 2017b) or built upon additional embeddings of the positions (Vaswani et al., 2017). On the other hand, when modeling orderless but relational data, the independence on positions makes self-attention a compelling choice. However, as far as we know, there is little work applying self-attention beyond the sequential data. In this work, we extend it to learn representations of multi-field categorical data in prediction tasks.

Taking the prediction of recommendation systems in e-commerce as an example, the service provider builds models to predict how much a customer (with features of gender, age[1], etc.) does like a product (with features of category, price, etc). The recommendations are then made by ranking the prediction scores. Quite often, input features are not independent, e.g., female customers are more prone to buying dresses than male customers. Combinations over the two fields, i.e., gender and category, can thus lead to more accurate predictions. In applications, categorical features are firstly transformed into sparse representations via one-hot encoding. The combinations in linear models are then made by cross product over different fields. Due to the sparsity problem, the combinations rely on much manual work of domain experts (Lian et al., 2018; Wang et al., 2017a; Qu et al., 2016).

Factorization based models instead additionally learn an embedding vector for each feature (Rendle, 2010). They compute interactions of features by their inner products. Compared with linear models,

---

[1]Note that numerical feature can be buketized to category

factorization based models can generalize well to unseen or rare features combinations. However, the simple inner product operation limits its approximation ability to some extent. DNN based models, further extend that by learning more complicated interactions. Typically, all embeddings are firstly concatenated into a long vector as a representation of all features and then fed into vanilla multi layer perceptions (MLPs). Recently, several works (Qu et al., 2016; Guo et al., 2017; Lian et al., 2018) improve the structure by adding product units (Durbin & Rumelhart, 1989) to construct explicit interactions. However, none of these works improve the representation rather than simple concatenation. In parallel, Liu et al. (2015) consider relating the embeddings of neighboring features with convolutional neural networks. However, it highly depends on the positions of the features. Different orders of features can yield significant different results while it is NP-hard to find the optimal (Chan et al., 2018). As introduced earlier, in this work, we instead apply self-attention to transform the feature embeddings due to it is insensitive to positions. The output vector in each position aggregates the input vector with its related feature embeddings. Thus we denote the outputs of self-attention as the representations of feature combinations. The contribution of this work is summarized as follows:

(a) Under the observation that self-attention models dependency without regard to positions, we extend it to model orderless while relational categorical features in prediction tasks. As far as we know, this is the first work of self-attention applying to non-sequential data.

(b) By utilizing self-attention, we are able to relate features in different fields and automatically learn representations of sophisticated feature combinations, which are known as the ingredient in many successful linear prediction models. In contrast, most of current DNN-based models simply concatenate all feature embeddings.

(c) In the original self-attention structure, each vector is aggregated with all vectors weighted by similarity. This reduce effective resolution and Multi-Head Attention is thus proposed. In this work, we further improve that by restricting the similarity weight to top-$k$ non-zero values. This modification is also consistent with the prior knowledge that one feature is almost impossible to be of relevance with all others. The truncation thus additionally works as a regularization (see Subsection 3.2).

(d) As a concrete example, we conduct experiments on CTR estimation. Across the two benchmark CTR datasets, i.e., Cretio and Avazu, our model with top-$k$ restricted self-attention achieves state of the art performance. Compared with vanilla DNN based model, the gain by adding self-attention is significantly larger than that by modifying the network structures, which most current works focus on. Besides, we give some interpretation on the combinations learned via self-attention.

## 2 Existing Embedded Based Models

The embedded based models for prediction over multi-field categorical data can mainly be divided into two categories: Factorization and DNN based models. The first work among factorization based models is Factorization Machine (FM)(Rendle, 2010). In addition to the linear part, it adds an additional interaction part for the prediction, i.e.,

$$f(\mathbf{x}) = b + \langle \mathbf{w}, \mathbf{x} \rangle + \sum_{i=1}^{n} \sum_{j=i+1}^{n} \langle \mathbf{e}_i, \mathbf{e}_j \rangle x_i x_j, \tag{1}$$

where $\mathbf{e}_i \in \mathbb{R}^k, k \ll n$ represents the low dimensional embedding vector for feature $i$. FM computes all possible second order interactions. For effective feature combinations that are rare or even unseen in the training data, FM can still generalize to the testing set. High order FM extends that by interacting with more features. As pointed out by Xiao et al. (2017), some interactions of FM could be noisy as these features may have no intrinsic relations. Xiao et al. (2017) instead learns an additional weight for each interaction with a neural network for further distinguishing its effectiveness. Juan et al. (2016) also extend FM to Field aware Factorization Machines (FFM). FFM learns different embeddings for each feature to be interacted with different fields, which achieves better performance at the cost of larger memories.

Most of DNN based models, instead, concatenate all looking up embeddings into a long vector, which is then fed into different neural networks. By using vanilla MLP, Zhang et al. (2016) propose

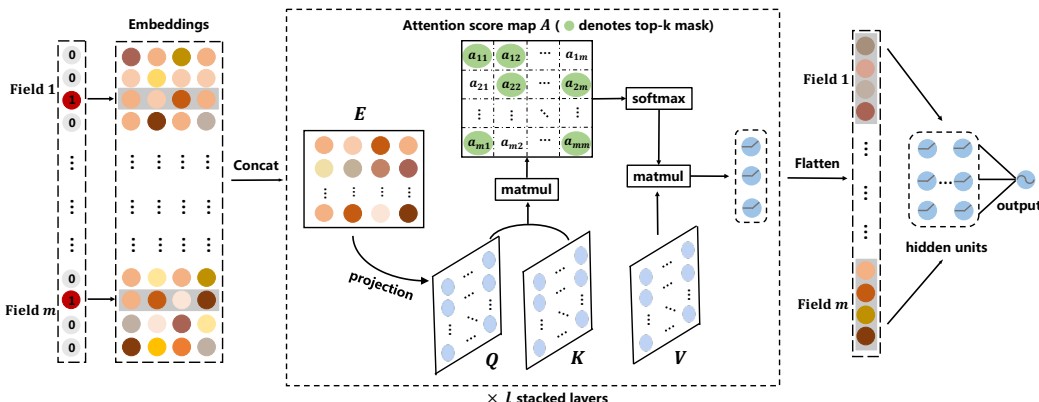

Figure 1: The overview of our model structure. For better illustration, we omit the skip connection in the self-attention and set the number of head to 1.

FNN to learn the implicit interactions. The embeddings are initialized with FM, which could also limit their performance. PNN (Qu et al., 2016) extends that by inserting additional product units (Durbin & Rumelhart, 1989) to the network. Although the product operation can be approximated arbitrarily well by a two layer's neural network with only 4 neurons in the hidden layer, see Theorem (Lin et al., 2017a), the extra units improve the performance. Recently, Guo et al. (2017) propose DeepFM by incorporating MLP with FM. To some extent, the structure can also be regarded as adding product units, but to the final layer instead of to the input layer as PNN. xDeepFM (Lian et al., 2018) further replaces the FM part by constructing higher order interactions. Although achieving better performance than vanilla MLP, all of these works build upon the simple concatenation, which could be insufficient to model the features relations.

## 3 MODEL STRUCTURE

In this section, we introduce our model for the prediction tasks. Our work builds upon the self-attention structure introduced by (Vaswani et al., 2017). Figure 1 gives an overview of the overall architecture. As shown in the left part of Figure 1, we firstly concatenate the embeddings as a matrix:

$$\boldsymbol{E} = [\ldots, \boldsymbol{e}_i, \ldots, \boldsymbol{e}_j, \ldots]^T, \tag{2}$$

where $\boldsymbol{E} \in \mathbb{R}^{m \times d}$, $d$ and $m$ denotes the dimension of the field embedding and the number of fields, respectively. Then the embeddings are transformed via stacked self-attention. After the transformation, the outputs are fed into the final neural network. Note that we use skip connections within self-attention, which guarantees that the model at least will not be worse than the basic DNN model without self-attention (He et al., 2016).

### 3.1 SELF-ATTENTION

Self-attention maps the input with weighted sum to the output by computing its similarity over different positions. It consists of three parts, i.e., the queries, the keys, and the values. All of them are derived from the same embedding matrix $\boldsymbol{E}$ by linear projection, usually following with the ReLU activation:

$$\boldsymbol{Q} = \max(\boldsymbol{E}\boldsymbol{W}_q, 0), \quad \boldsymbol{K} = \max(\boldsymbol{E}\boldsymbol{W}_k, 0), \text{ and } \boldsymbol{V} = \max(\boldsymbol{E}\boldsymbol{W}_v, 0), \tag{3}$$

with $\boldsymbol{W}_q \in \mathbb{R}^{d \times d}$, $\boldsymbol{W}_k \in \mathbb{R}^{d \times d}$, and $\boldsymbol{W}_v \in \mathbb{R}^{d \times d}$ being learned parameters. We then compute dot products of the queries with the keys. The resulting matrix is denoted as the attention score map, i.e., the $A$ in upper Figure 1. The element $a_{ij}$ captures the relation between the $i$-th position and $j$-th position . By applying a softmax function to the score map, we obtain the matrix of outputs:

$$\tilde{\boldsymbol{V}} = \text{softmax}\left(\frac{\boldsymbol{Q}\boldsymbol{K}^T}{\sqrt{d}}\right)\boldsymbol{V} + \boldsymbol{E}, \tag{4}$$

where $\sqrt{d}$ normalizes the weights. Additionally adding $\boldsymbol{E}$ denotes the skip connection. And the softmax operates on each row, ensuring all the computed weights summing up to one. $\tilde{\boldsymbol{V}}$ is further transformed by a two layers' network with ReLU activation:

$$\tilde{\boldsymbol{E}} = \max(\tilde{\boldsymbol{V}}\boldsymbol{W}_1, 0)\boldsymbol{W}_2 + \tilde{\boldsymbol{V}}, \tag{5}$$

where the latter part also represents the skip connection.

In equation 4, each row of $\tilde{\boldsymbol{V}}$ aggregates all values of $V$ with weighted sum. This operation reduces the effective resolution of attention especially for the data with many fields. To alleviate it, the multi-head structure is proposed in the original paper (Vaswani et al., 2017). By projecting the embedding $E$ into $h$ different sets of queries, keys, and values, multi-head attention computes the aggregation as

$$\tilde{\boldsymbol{V}} = \left[\text{softmax}\left(\frac{\boldsymbol{Q}_1\boldsymbol{K}_1^T}{\sqrt{d/h}}\right)\boldsymbol{V}_1, \cdots, \text{softmax}\left(\frac{\boldsymbol{Q}_h\boldsymbol{K}_h^T}{\sqrt{d/h}}\right)\boldsymbol{V}_h\right] + \boldsymbol{E}, \tag{6}$$

which is then processed by equation 5. We also stack several layers of self-attention for further performance improvements.

## 3.2 TOP-$k$ SPARSITY

The multi-head attention structure alleviates the resolution problem by operating on several subspaces. In this subsection, we further improve the structure by ensuring sparsity on the weights after the softmax function. Intuitively, each feature is impossible to have relations with all others. Weighting on all the values as equation 4 will include both useful and useless combinations. The useless feature combination may introduce additional noise and adversely harm the performance. Instead, by ensuring sparsity on the weights, we focus on more related features. Besides, the sparsity constraint will filter out some unrelated connections, which to some extent regularize the model.

To ensure the sparsity, a feasible solution is to directly set a threshold. The element of the attention score map $\boldsymbol{A}$ is set to $-\infty$ when it is less than the threshold, and is preserved when greater than or equal to the threshold. However, in practice, we find that it is not easy to choose a proper threshold. Instead, we adopt the simple top-$k$ restriction, i.e., preserving the top-$k$ values in each row of $\boldsymbol{A}$ and set the rest to $-\infty$. Then equation 4 becomes

$$\tilde{\boldsymbol{V}} = \text{softmax}\left(\frac{\text{TopK}(\boldsymbol{Q}\boldsymbol{K}^T, k)}{\sqrt{d}}\right)\boldsymbol{V} + \boldsymbol{E}, \tag{7}$$

where the TopK$(\cdot, \cdot)$ function is define as

$$\text{TopK}(\boldsymbol{A}, k)_{ij} = \begin{cases} a_{ij}, & \text{if } a_{ij} \text{ is the top } k \text{ elements in the } i\text{-th row of } \boldsymbol{A}, \\ -\infty, & \text{otherwise.} \end{cases} \tag{8}$$

The top-$k$ restriction is also used in (Shazeer et al., 2017) to threshold the value in the softmax, where it is mainly for reducing computation of each expert network. Here we are to improve the resolution of averaging and further regularize the model. In practice, it is used in conjugate with multi-head structure. As the experiments shows, it can additionally improve the performance.

## 4 EXPERIMENTS

In this section, we conduct experiments to validate the effectiveness of adding self-attention. As a concrete example, we test on the click trough rate (CTR) prediction task, i.e., to estimate the probability that a user would like to click a given item. CTR prediction is at the core position of the commercial recommendation systems and on-line advertising. In recommendation systems, CTR is often used to rank all candidate items. In on-line advertising, the revenue of per click is measured by CTR $\times$ bid price, where accurate CTR prediction is the essence. The two most common evaluation metrics of CTR are area under the curve (AUC) and Logloss:

$$\text{Logloss} = -\frac{1}{N}\sum_{i=1}^{N}\left(y_i\log(p_i) + (1 - y_i)\log(1 - p_i)\right), \tag{9}$$

where $N$ is the number of samples, $y_i$ is the 0/1 label, and $p_i$ is the estimated probability.

Table 1: Testing results on the Criteo dataset. We conduct two versions of our Attention-Net, i.e., with and without using the top-$k$ restriction. The largest AUC and the least Logloss are in bold.

| Model | Criteo | | | |
|---|---|---|---|---|
| | AUC (%) | Relative | Logloss ($\times 10^{-2}$) | Relative |
| FM (Rendle, 2010) | 79.53 | $-0.79$ | 45.47 | $+0.84$ |
| FFM (Juan et al., 2016) | 79.84 | $-0.48$ | 45.09 | $+0.46$ |
| AFM (Xiao et al., 2017) | 80.01 | $-0.31$ | 44.68 | $+0.05$ |
| Vanilla MLP | 80.32 | 0 | 44.63 | 0 |
| CCPM (Liu et al., 2015) | 80.17 | $-0.15$ | 44.83 | $+0.20$ |
| PNN (Qu et al., 2016) | 80.47 | $+0.15$ | 44.55 | $-0.08$ |
| DeepFM (Guo et al., 2017) | 80.37 | $+0.05$ | 44.59 | $-0.04$ |
| xDeepFM (Lian et al., 2018) | 80.69 | $+0.37$ | 44.29 | $-0.34$ |
| Attention-Net (without top-$k$) | 81.17 | $+0.85$ | 44.05 | $-0.58$ |
| Attention-Net (with top-$k$) | **81.38** | **$+1.06$** | **43.91** | **$-0.72$** |

## 4.1 EXPERIMENT SETUP

We compare with existing state of art embedded based models, i.e., FM (Rendle, 2010), FFM (Juan et al., 2016), AFM (Xiao et al., 2017), vanilla MLP, PNN (Qu et al., 2016), DeepFM (Guo et al., 2017), and xDeepFM (Lian et al., 2018). All codes are run in Tensorflow (Abadi et al., 2016). For simplicity, we use the inner product version of PNN. The embeddings are initialized with i.i.d. samples from Gaussian distribution with standard deviation 0.001. We adopt Adam (Kingma & Ba, 2015) as the optimizer with learning rate 0.001 and mini-batch size 1024. The gradient norm is clipped in the range $[-5, 5]$. For DNN based models, we use ReLU as the activation function and sigmoid in the final layer. As we are to test effectiveness of self-attention, here we simply use vanilla MLP in the final prediction part. Note that the attention structure is general enough for existing work to build upon. We denote our models as Attention-Net. The hyper-parameter $k$ is set to 5.

## 4.2 CRETIO DATASET

We first conduct the experiments on the CTR benchmark dataset Criteo[2]. The dataset includes 45 million users' click records, where 13 fields of the features are continuous and the rest 26 fields are in category. For continuous features, we firstly compute $\log(\cdot)$ on the values, which are then bucketized with boundaries defined as continuous integers. For categorical features, we truncate the number of categories to 5000 with all low frequency features as the "unknown" category. As the testing set is not available, we follow the same experiment setting as (Guo et al., 2017), that is, splitting the dataset into two parts: 90% is for training, while the rest 10% is for testing. For vanilla MLP, PNN, DeepFM, xDeepFM, and our Attention-Net, the common part MLP is set to three layers with the number of hidden layer neurons to be 600 and 400, respectively. All methods are terminated after 120k training steps.

The testing results of all the compared methods are shown in Table 1. For the ease of comparison, we denote the vanilla MLP as the baseline. Note that an improvement of 0.1% in AUC is usually regarded significant for the CTR prediction. Among all the compared methods, FM performs the worst, which is mainly because that it constructs all possible interactions, in which some could be very useless. AFM improves FM by learning an additional weights for each interaction. Overall, the factorization based models are worse than the DNN based models. The intrinsic second order interactions limit their ability to learn more sophisticated feature combinations. Among all DNN based models, CCPM gives the lowest AUC. It is possible for CCPM to achieve better performance by modifying the relative positions of the features. However, this would need much extra work, especially when the meaning of all features is unknown, e.g., in this Cretio dataset. By adding additional product units to the network, PNN, DeepFM, and xDeepFM all surpass the vanilla MLP.

---

[2]http://labs.criteo.com/2014/02/download-kaggle-display-advertising-challenge-dataset/

Table 2: Testing results on the Avzau dataset.

| Model | Avzau | | | |
|---|---|---|---|---|
| | AUC (%) | Relative | Logloss ($\times 10^{-2}$) | Relative |
| FM (Rendle, 2010) | 78.01 | $-0.35$ | 37.99 | $+0.20$ |
| FFM (Juan et al., 2016) | 78.29 | $-0.07$ | 37.83 | $+0.04$ |
| AFM (Xiao et al., 2017) | 78.19 | $-0.17$ | 37.92 | $+0.13$ |
| Vanilla MLP | 78.36 | 0 | 37.79 | 0 |
| CCPM (Liu et al., 2015) | 78.05 | $-0.31$ | 38.01 | $+0.22$ |
| PNN (Qu et al., 2016) | 78.41 | $+0.05$ | 37.80 | $+0.01$ |
| DeepFM (Guo et al., 2017) | 78.47 | $+0.11$ | 37.65 | $-0.14$ |
| xDeepFM (Lian et al., 2018) | 78.67 | $+0.31$ | 37.55 | $-0.24$ |
| Attention-Net (without top-$k$) | 78.85 | $+0.49$ | 37.36 | $-0.43$ |
| Attention-Net (with top-$k$) | **79.21** | **$+0.85$** | **37.17** | **$-0.62$** |

Among all the compared methods, our Attention-Net achieves the most plausible performance. Our model with the top-$k$ restriction outperforms the vanilla MLP by nearly 1% in the term of AUC. The improvement is several times larger than the works by adding product units, i.e., PNN, DeepFM, and xDeepFM. Note that the only difference between our Attention-Net and the vanilla MLP is the additional attention mechanism. By using self-attention, we are able to learn representations of feature combinations, which as input are more suitable than simple concatenation for the final prediction. Considering the last two rows of Table 1, we conduct two versions of our Attention-Net. With the top-$k$ restricted self-attention, our model can attain additional 0.21% AUC increase. As discussed earlier, by ensuring sparsity on the weight matrix, we could increase the effect resolutions when averaging over different fields. Besides, the sparsity regularizes the model by removing useless combinations.

### 4.3 Avazu Dataset

We also conduct experiments on the Avazu dataset[3]. It includes 40 million users' click records, where all the 22 fields are in category. We also preprocess the categorical features as done in the Cretio dataset. Following (Qu et al., 2018), we randomly split the dataset into training and test sets at $4:1$. For vanilla MLP, PNN, DeepFM, xDeepFM, and our Attention-Net, the common MLP part is set to three layers with the number of hidden layer neurons to be 400 and 200, respectively. The results are shown in Table 2, which is in consistent with results of the Cretio dataset. Among all compared methods, our Attention-Net achieves the most plausible performance. The top-$k$ restriction itself can lead to 0.36% increase in the term of AUC. All methods are terminated after 135k training steps.

### 4.4 Hyper-Parameter Study

To evaluate the importance of different components of the Attention-Net, we varied our base model in different ways, including (1) the number of attention layers $l$; (2) the dimension of embeddings $d$; (3) the number of attention heads $h$; (4) the hyper-parameter $k$; (5) the regularization parameter $\lambda$; and (6) the structure of MLP. The thorough comparisons are presented in Table 3.

In Table 3 rows (A), we observe that, the stacked attention layers bring an improvement of 0.2% to AUC when $L$ varies from 1 to 3, which degrades beyond that. This is a common trend in machine learning as network training becomes more difficult with increasing depth. In rows (B), we see a substantial improvement as the dimension is expanded from 8 to 48. Due to the increasing burden of optimizing MLP part, as expected, performance also degrades with higher dimensionality.

In Table 3 rows (C), we show the effect of varying the number of heads on the final performance. When $h$ is 1, the compatibility function for attention resembles a simple dot product, which leads to a reduction of about 0.45% on the results. This indicates a fine-grained attention scoring function with

---

[3]http://www.kaggle.com/c/avazu-ctr-prediction

Table 3: Variations on the Attention-Net. Unlisted values are identical to those of the base model. All metrics are evaluated on the Avzau dataset. $l$, $d$, $h$, $k$, and $\lambda$ denote self-attention layers, embedding dimension, heads, top-$k$, and the regularization parameter, respectively.

| | $l$ | $d$ | $h$ | $k$ | $\lambda$ | Neurons | AUC (%) | Logloss ($\times 10^{-2}$) |
|---|---|---|---|---|---|---|---|---|
| base | 3 | 32 | 4 | 5 | 0 | $400, 200$ | 79.21 | 37.17 |
| (A) | 1 | | | | | | −0.22 | +0.07 |
| | 2 | | | | | | −0.06 | +0.02 |
| | 4 | | | | | | −0.10 | +0.02 |
| (B) | | 8 | | | | | −0.22 | +0.15 |
| | | 16 | | | | | −0.04 | +0.07 |
| | | 48 | | | | | +0.07 | +0.02 |
| | | 64 | | | | | −0.13 | +0.17 |
| (C) | | | 1 | | | | −0.45 | +0.31 |
| | | | 2 | | | | −0.32 | +0.21 |
| | | | 8 | | | | −0.10 | +0.09 |
| (D) | | | | 2 | | | −0.64 | +0.39 |
| | | | | 10 | | | −0.05 | +0.01 |
| | | | | 22 | | | −0.36 | +0.19 |
| (E) | | | | | 1e-6 | | −0.14 | +0.09 |
| | | | | | 1e-5 | | −0.24 | +0.19 |
| | | | | | 1e-4 | | −0.52 | +0.41 |
| (F) | | | | | | $200, 200$ | −0.21 | +0.12 |
| | | | | | | $400, 400$ | +0.02 | +0.01 |
| | | | | | | $600, 400$ | +0.05 | −0.01 |

different heads has the advantage of learning sophisticated feature interactions. We see a steady rise in AUC as the number of heads is increased from 1 to 4, which drops off beyond that.

In Table 3 rows (D), we show the effect of varying $k$. Given that the number of fields are 22, we set $k$ to be 2, 5, 10 and 22. The result demonstrates that the model achieves a remarkable improvement of 0.64% on AUC as $k$ is increased from 2 to 5. It justifies our motivation to learn representations of combinations. However, the performance deteriorates when further enlarging $k$. A steep drop is observed by removing top-$k$ strategy. We conclude that the top-$k$ restriction can provide a boost by improving the resolution effectively. In contrast to traditional 'plain' attention, it yields field-aware attention masks which regulate the flow of information across multi-field categorical features.

We further observe in rows (E) and rows (F) that, performance increases with more hidden units of MLP, whereas adding the $L_2$ regularization has the opposite effect.

## 4.5 ILLUSTRATION OF FEATURE COMBINATIONS

In this subsection, we give an interpretation of feature combinations learned by self-attention. For the concern of privacy, each feature in the above two datasets is cryptographic. The only public information is the name info of 12 fields among all 22 in the Avazu dataset. Thus it is meaningless to visualize the attention weights as done in (Vaswani et al., 2017). Instead, here we utilize the tool of information theory for feature selection. In information theory, it defines three types of relation, i.e., relevance, redundancy, and complementarity (Meyer et al., 2008). In this work, we are in most concern of complementarity. The complementarity between two random features $x_i$ and $x_j$ and the output $y$ is defined as

$$\boldsymbol{C}(x_i, x_j) = \boldsymbol{I}(\{x_i, x_j\}; y) - \boldsymbol{I}(\mathbf{x}_i; y) - \boldsymbol{I}(\mathbf{x}_j; y) \tag{10}$$

where $\{x_i, x_j\}$ denotes the combination of two features and $\boldsymbol{I}(x; y)$ is defined as

$$\boldsymbol{I}(x; y) = \boldsymbol{H}(x) + \boldsymbol{H}(y) - \boldsymbol{H}(x, y), \tag{11}$$

with $\boldsymbol{H}(\cdot)$ computing the entropy. $\boldsymbol{C}(\cdot, \cdot)$ measures, in bits, the gain resulting from using the joint mutual information of two features and instead of the sum of the univariate informations. A well-

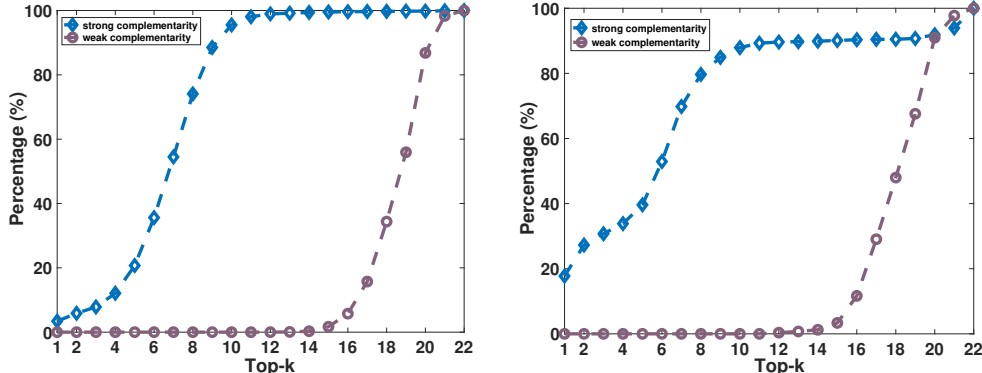

Figure 2: Percentage accumulation of different feature combinations that arise in the top-$k$ of attention score map $\boldsymbol{A}$. For each of the two experiments, we randomly choose two combinations of features with relative strong and weak complementarity. The detailed description of all chosen features is listed in Table 4, where the left sub-figure corresponds to the upper part.

Table 4: Description of features used in Figure 2. As all features are cryptographic, we only know the name of the field and the hash value of each feature. We measure the corresponding CTR and compute the complementarity for the combinations.

| Feature $x_i$ | Feature $x_j$ | CTR (%) $x_i, x_j, \{x_i, x_j\}$ | $\boldsymbol{C}(\mathbf{x}_i, \mathbf{x}_j)$ |
|---|---|---|---|
| app_category 07d7df22 | site_domain c4e18dd6 | (19.93, 12.20, 23.57) | 0.2633 |
| | site_id 6256f5b4 | (19.3, 18.24, 20.33) | 0.0433 |
| app_domain 7801e8d9 | banner_pos 0 | (19.50, 16.40, 22.13) | 0.1704 |
| | site_id 6256f5b4 | (19.50, 18.24, 19.90) | 0.0542 |

known illustration of complementarity is XOR problem. The individual input $x_i$ and $x_j$ have a null relevance, i.e., $\boldsymbol{I}(\mathbf{x}_i; y) = \boldsymbol{I}(\mathbf{x}_j; y) = 0$. While the combination $\{x_i, x_j\}$ has the maximal relevance, $\boldsymbol{I}(\{x_i, x_j\}; y) = \boldsymbol{H}(y) > 0$.

Now we are to examine whether self-attention could combine features of strong complementarity. We randomly pick feature $x_i$ and compute $\boldsymbol{C}(x_i, \cdot)$ for all feasible combinations. We choose two combinations with relatively strong and weak complementarity. Then we count the occurrence percentage that the score $a_{ij}$ is in the top-$k$ of attention score map $\boldsymbol{A}_{i,:}$. Figure 2 plots the accumulation curves for two sets of experiments. Constantly, the combinations with strong complementarity are more prone to be combined by self-attention. While the combinations with very weak complementarity are less prone to be combined. We also give detailed description of all the combined features in Table 4.

## 5 CONCLUSION

In this work, we extend self-attention to model orderless while relational categorical data in prediction tasks. By utilizing self-attention to transform the input embedding, we can relate feature in different fields and automatically learns representations of feature combinations. In contrast, most existing works simply concatenate all embeddings together. We further modify the original self-attention structure by ensuring sparsity on the weight matrix, which not only increases the effective resolution of aggregating values, but also regularizes the model by removing useless while noisy combinations. As a concrete examples, we conduct experiments on the CTR prediction. Across two benchmark datasets, our Attention-Net with the top-$k$ restriction consistently achieves the most plausible results. We also give an illustration on the learned feature combinations.

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
