# OpenReview forum: "Learning Representations of Categorical Feature Combinations via Self-Attention"
_ICLR.cc/2019/Conference_

### Official Review · AnonReviewer1 · 2018-11-02
**Concern of invalid evaluation and a weak contribution**

**Rating:** 5
**Confidence:** 4

**Review:**

Quality:
- In 4.4, the authors have vigorously explored the space of hyperparameters. However, they do not describe how to determine the hyperparameters, e.g., set aside a validation set from a part of the training set and determine the hyperparameters using this validation set, while the authors split the two datasets into only training and test sets, respectively. Without this procedure, the results may overfit to the test set via repeated experiments. Even though the used datasets are of few-million, this procedure guarantees a minimum requirement for a reliable outcome from the proposed model. I firmly recommend the authors to update their results using a validation set to determine the hyperparameters and then report on the test set. Please describe these experimental details to ensure that the performed experiments are valid.

Clarity:
- Overall, the writing can be improved via proof-reading and polishing the sentences. In Introduction section, "there is little work applying..." can be specified or rephrased with "it is underexplored to apply", and "input features are not independent" can be specified on what there are not independent. Moreover, the last two sentences in the second paragraph in the Introduction section is unclear what the authors want to argue: "The combinations in linear models are then made by cross product over different fields. Due to the sparsity problem, the combinations rely on much manual work of domain experts."
- The authors use top-k restriction (Shazeer et al., 2017) to consider sparse relationships among the features. For this reason, have you tried to use the L1 loss on the probability distributions, which are the outputs of softmax function?
- In 4.5, the authors said, they "are in most concern of complementarity." What is the reason for this idea and why not the "relevance"?
- In Table 4, I'm afraid that I don't understand the content (three numbers in parenthesis) of the third column. How does each input x_i or x_j, or a tuple of them get their own CTR?

Originality and significance:
- They apply self-attention to learn multiple categorical features to predict Click-Through-Rate (CTR) with a top-k non-zero similarity weight constraint to adapt to their categorical inputs. Due to this, the scientific contribution to the corresponding community is highly limited to providing empirical results on the CTR task.
- The authors argue that "most of current DNN-based models simply concatenate all feature embeddings"; however, this argument might be an over-simplified statement for the existing models in section 2.
- Similar works can be found but missed to cite: [1] proposes a general framework to self-attention to exploit sequential (time-domain) and parallel (feature-domain) non-locality. [2] learns bilinear attention maps to integrate multimodal inputs using skip-connections and multiple layers on top of the idea of low-rank bilinear pooling.

Pros:
- Strong empirical results on two CTR tasks using the previous works of self-attention and top-k restriction techniques.

Cons:
- This work fairly lacks its originality since the proposing method heavily relies on the two previous works, self-attention and top-k restriction. They apply them to multiple categorical features to estimate CTR; however, their application seems to be monotonic without a novel idea of task-specific adaptation.

Minor comments:
- In Figure 1, "the number of head" -> "the number of heads".


[1] Wang, X., Girshick, R., Gupta, A., & He, K. (2018). Non-local Neural Networks. In IEEE Computer Society Conference on Computer Vision and Pattern Recognition (CVPR'18).
[2] Kim, J.-H., Jun, J., & Zhang, B.-T. (2018). Bilinear Attention Networks. In Advances in Neural Information Processing Systems 32 (NIPS'18).

---

### Official Review · AnonReviewer2 · 2018-11-02

**Rating:** 5
**Confidence:** 3

**Review:**

Summary:
The authors apply the self-attention mechanism, a.k.a. transformer, to improve the representations of multi-field categorical features in recommendation systems. Unlike the previous approaches in which multi-field features are simply concatenated, the proposed method more actively combines those features improving the final performance.

Strengths:
+ It is reasonable to apply the permutation-invariant self-attention mechanism to the multi-field features as orders of the fields should not matter.
+ The method achieves the state-of-the-art performance on two datasets.

Weaknesses:
- The paper lacks the technical novelty as it does not propose any novel technique. Rather, it simply applies an existing technique to a new type of dataset.
- More extensive analyses on the learned representation would improve the paper.
- As the authors argue, the method can be used upon other existing state-of-the-art networks. Showing the improvement on other methods would improve the paper. Currently, the authors only present improvement on a simple MLP.

Questions:
To apply the self-attention, the embeddings of the field features should be projected in the same space. I wonder if this physically makes sense. I wonder how they are embedded in the features and relate to each other. I would suggest to include some analysis on the features while putting some rows of Table 3 to the appendix since many of these rows are not directly related to the method itself.

Overall, I like the idea of the paper. However, the paper lacks the technical novelty and presents only limited experiments and analysis. I would suggest the authors include more analyses on the learned representations.

---

### Official Review · AnonReviewer3 · 2018-11-03
**Good results, better justification for the novelty needed.**

**Rating:** 5
**Confidence:** 4

**Review:**

Summary
The paper proposes to apply self-attention mechanism from (Vaswani et.al.) to the task of click-through rate prediction, which is a task where one has input features which are a concatenation of multiple one-hot vectors (referred to as fields). The paper finds that applying the self-attention mechanism outperforms state of the art approaches for the task on two benchmark datasets. It then proposes a small modification to the self-attention mechanism, retaining only the top-k attentions to sparsify attention, and finds that it leads to marginal improvements.

Strengths
+ The paper is fairly well written, and the contributions are succinctly summarized.
+ The proposed approach appears to get state of the art results on click-through rate prediction.
+ The results contain clear ablations of the approach.

Negatives
1. It is not clear why the skip connection is needed. Especially, using the skip connection the way it is done in Eqn. 4 is a bit odd since we are adding positive quantities to each other, meaning that across multiple rounds, the magnitude of the attended feature will keep increasing. Perhaps this is the reason why performance deteriorates after attending thrice?

2. Calling top-K a regularizer is somewhat misleading as it is a fundamentally different model class, as opposed to a regularizer that imposes a soft constraint on the kind of solutions that should be preferred in our hypothesis class. The current paper does not show with enough clarity if the improvements with top-k are because it is a better model for the data or because it is a better regularizer. One way to do this would be to systematically look at the difference between training and validation losses with and without top-k and show that the difference is smaller when the model is regularized.
More generally, it would be ideal to show what kind of a constraint the top-k attention places on the hypothesis class of the original model. For example, the dropout paper shows that dropout, in the linear case is equivalent to L2 regularization (in expectation). (*)

3. It would be interesting to report how often there is an overlap in the top k indices chosen across multi-head attentions.

4. What are the relative number of parameters in each of the models for which the results are reported? Are we ensuring that a similar number of parameters are used to report all the results in say, Table. 1.? Also, it would be good to report error bars for the results in Table. 1 since the differences seem to quite small. (*)


Preliminary Evaluation
The paper is a fairly straightforward application of self-attention to the task of click-through rate prediction. The major modeling novelty is in using top-k attentions for the click-through task, the interestingness/ validity of which needs to be demonstrated more clearly to understand if this heuristic might apply to other models and other datasets. Important points for the rebuttal are marked with a (*) above.

---

### Meta-Review · Area_Chair1 · 2018-12-02
**Reject**

**Confidence:** 4
**Recommendation:** Reject

**Metareview:**

All reviewers agree in their assessment that this paper is not ready for acceptance into ICLR and the authors did not respond during the rebuttal phase.